# The Effect of Different Surface Conditioning Techniques on the Bonding between Resin Cement & Ceramic

Shekhar Gupta [1] , Bharti Gupta [2], Bhagwandas K. Motwani [3], Sultan Binalrimal [4] , Waseem Radwan [4],
Ali Robaian [5], Bassam Zidane [6] , Mohammed Hussain Dafer Al Wadei [7], Vishnu Priya Veeraraghavan [8],
Shilpa Bhandi [9], A. Thirumal Raj [10] and Shankargouda Patil [2,*]

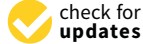



1 Department of Prosthetic Dental Science, College of Dentistry, Jazan University, Jazan 45142, Saudi Arabia;
drshekhar786@gmail.com
2 Department of Maxillofacial Surgery and Diagnostic Sciences, Division of Oral Pathology,
College of Dentistry, Jazan University, Jazan 45142, Saudi Arabia; drbhartisaraf@gmail.com
3 Department of Prosthodontics, Swargiya Dadasaheb Kalmegh Smruti Dental College & Hospital,
Nagpur 441110, India; motwanimukta@yahoo.co.in
4 Department of Restorative Science, College of Dentistry, Riyadh Elm University, Riyadh 12611, Saudi Arabia;
Sultan@riyadh.edu.sa (S.B.); waseem.radwan@riyadh.edu.sa (W.R.)
5 Department of Conservative Dental Sciences, College of Dentistry, Prince Sattam bin Abdulaziz University,
Alkharj 16273, Saudi Arabia; ali.alQahtani@psau.edu.sa
6 Department of Restoratives Science, King Abdulaziz University, Jeddah 21589, Saudi Arabia;
bzidane@kau.edu.sa
7 Department of Restorative Dental Science, King Khalid University, Abha 61421, Saudi Arabia;
moalwadai@kku.edu.sa
8 Centre of Molecular Medicine and Diagnostics (COMManD), Department of Biochemistry, Saveetha Dental
College & Hospitals, Saveetha Institute of Medical and Technical Sciences, Saveetha University,
Chennai 600077, India; drvishnupriyav@gmail.com
9 Department of Restorative Dental Sciences, College of Dentistry, Jazan University, Jazan 45142, Saudi Arabia;
shilpa.bhandi@gmail.com
10 Department of Oral Pathology and Microbiology, Sri Venkateswara Dental College and Hospital,
Chennai 600130, India; thirumalraj666@gmail.com
* Correspondence: dr.ravipatil@gmail.com

**Abstract:** Surface etching before cementation is a vital step that determines the clinical performance
of ceramic restorations. Etching alters surface topography that contributes effective bonding between
ceramic restoration and resin cement. This study aimed to compare etching techniques to determine
the most effective etching method contributing the highest bond strength that helps in improving
dental implants. Materials and methods: sixty discs of feldspathic ceramic measuring 10 mm diameter
and 4 mm thickness were prepared. The 60 samples were divided into four equal groups based on the
surface treatment technique used: group A: 9.6% hydrofluoric acid; group B: coarse diamond burs;
group C: $CO_2$ laser; and group D: no treatment. Ceramic disc specimens were examined under a
Scanning Electron microscope (SEM) after surface treatment to characterize their surface morphology.
Further, the specimens were luted with a resin luting agent and incubated for 24 h at a temperature
of 37 °C simulating the oral environment. After 24 h, shear bond strength (SBS) and the nature of
bond failure was determined for each specimen using a universal Instron testing machine. Results:
significant change in surface morphology was noticed on hydrofluoric acid treatment forming larger
irregular roughness (4.83 ± 1.78 μm) with multiple patterns of grooves and pores compared to other
groups. Further, the highest SBS value was measured on hydrofluoric acid etching that display the
highest bond strength due to the high surface roughness. In conclusion, our findings report a strong
association between the surface roughness and bond strength upon hydrofluoric acid compared to
other methods. Further work in this direction will enhance the utility of the etching technique on the
improvement of dental implants.

**Keywords:** ceramic; diamond burs; hydrofluoric acid; lasers; porcelain; shear bond strength; sur-
face treatment

## 1. Introduction

Dental porcelains are ideally suited for anterior esthetic restorations. Their color stability, resistance to staining, wear-resistance, and biocompatibility allow it to simulate natural teeth structure [1]. Their favourable esthetics combined with a conservative tooth preparation make them popular among patients and clinicians [2].

Feldspathic ceramics, made from silica, are a commonly used ceramic material in aesthetic restorative dentistry as a part of inlays, onlays, and veneers. Ceramics require a durable bond to attach to the natural tooth structure. Adhesion between dental ceramics and resins is predicated on the physio-chemical interaction between the adhesive and ceramic [3]. This requires pre-conditioning of the ceramic surface for optimum bonding prior to cementation with a resin [4]. Conditioning and preparing the ceramic involves etching with hydrofluoric acid. The duration of etching and concentration of the acid can affect the final adhesive bond strength. The two main interfaces that bonding occurs at are the ceramic-cement interface and the adhesive-dentin interface [5]. Bond strength is determined by these two interfaces [6]. Clinical failures of ceramic restorations predominantly emanate from cemented surfaces [7–9]. The integrity of the conditioned surface and the resin affects treatment success and the longevity of the restoration [10,11]. Surface treatments of ceramics are a fundamental necessity for bonding and influence the appearance of the restoration and even microbial retention [12].

There are several techniques to precondition ceramic surfaces to enhance bond strength, including chemical, mechanical, and laser irradiation [4,10,13]. Hydrofluoric acid causes topographical changes including surface dissolutions, which allows for micromechanical retention [13]. Etching using hydrofluoric acid is a popular method for conditioning surfaces. However, it often requires an additional step of the application of a silane coupling agent for adequate strength [14,15]. Hydrofluoric acid is hazardous to soft tissues [16]. Exposure can lead to irreparable damage to the eyes, and risks, such as necrosis and bone decalcification, make it a precarious substance to handle and employ [17,18]. Such concerns necessitate the exploration of different etching protocols which provide reliable long-term bonding to ceramics as alternatives to hydrofluoric acid [19,20].

Mechanical methods of surface conditioning involve air abrasion using aluminum oxide particles or diamond burs to roughen the ceramic surface [21]. Roughening with burs at high speed produces a reliably retentive surface. Though diamond bur roughening in combination with other surface modification methods may yield better bond strength, they can cause a reduction in the strength of the restoration [22]. Burs have an added risk of crack formation, which may propagate over time, leading to the failure of a restoration [23,24].

Laser irradiation is a modern method of conditioning ceramic surfaces to ensure adhesion. Er:YAG, carbon dioxide ($CO_2$), and Nd:YAG lasers can be used to condition the surfaces of ceramics prior to cementation. Lasers accumulate and focus an enormous amount of energy to a small circumscribed target area, which absorbs the energy and causes mircromorphological physical changes. The carbon dioxide laser ($CO_2$) is commonly used intraorally in soft tissue and hard tissue applications [25,26]. Dental porcelain approximately absorbs $CO_2$ laser wavelength. $CO_2$ laser is very suitable for the surface treatment of ceramic materials [27]. $CO_2$ laser etching may represent an effective method for conditioning zirconia surfaces, enhancing micromechanical retention, and improving the bond strength [28]. Conchoidal tears appear as a result of surface warming because of a heat initiation of ceramic surfaces by the focusing of the $CO_2$ laser, which may lead to mechanical success between resin composite and ceramics retention [26–30]. Parameters of wavelength, frequency of laser, and output power dictate the characteristics of the irradiated surface [30,31]. $CO_2$ lasers can produce sufficient bond strength in zirconia ceramics [32]. They have also been shown to increase surface roughness and wettability for certain CAD/CAM ceramics [33]. Lasers roughen the surface of feldspathic ceramic, enhancing micromechanical retention and improving the bond strength [30–32]. A few comparative studies have examined the laser treatment of feldspathic ceramic masses,

leading to a lack of consensus on the best method to produce optimal bond strength in feldspathic ceramics [15,30–38].

Herein, we hypothesize that the conditioning treatments may influence the surface microstructure on ceramic and, thereby, have an impact on bond strength. On the other hand, characterization of the surface microstructure of the conditioned ceramic would augment our understanding of the influence of conditioning treatments. Considering the lack of clinical evidence for alternatives to hydrofluoric acid for conditioning ceramics, the present study evaluated the effect of three different surface conditioning techniques on the bonding between dual-cure resin luting cement and surface-treated feldspathic ceramic specifically focus surfaces assessment with scanning electron microscope, shear bond strength, and nature of bond failure.

## 2. Materials and Methods

### 2.1. Ceramic Disc Fabrication and Test Specimens Grouping

Feldspathic ceramic (CERAMCO 3 (Dentsply, Burlington, NJ, USA)), a mixture of potassium feldspar, and glass is in the feldspathic porcelains. After incongruent melting, feldspathic porcelains contain 19 weight percentage (wt%) of leucite crystals ($K_2O \cdot Al_2O_3 \cdot 4SiO_2$). Pre weighted (0.7 gm) low fusing feldspathic ceramic powder was mixed with 200 μL of modeling liquid to form a slurry. The slurry was packed in the custom-made metallic mold and gently tapped on a vibrator to ensure no voids were formed. Blotting paper was used to remove the excess water. The disc was then retrieved with gentle pressure and placed on a custom-made phosphate bonded slab for firing. Disc specimens were fired in a Touch & Press Furnace (Dentsply Sirona, Charlotte, NC, USA) according to manufacturers' instructions. Initially, specimens were preheated for 5 min from 0 to 650 °C and then the temperature was maintained at 650 °C for another 5 min. The temperature was raised from 650 °C to 930 °C for 5 min at 50 hPa, followed by a vacuum hold for 1 min at 930 °C at 984 hPa. The vacuum was released to complete the firing cycle. Sixty ceramic test specimens with 4 mm thickness and 10 mm diameter were fabricated (Figure 1). These 60 samples of discs were divided into four groups:

- Group A—Treated with 9.6% hydrofluoric acid
- Group B—Treated with coarse diamond burs
- Group C—Treated with $CO_2$ laser
- Group D—Control group without any surface treatment.

### 2.2. Treating the Surface of Test Specimens

Surface treatment was applied to the self-glazed surface of the prepared ceramic discs. The ceramic discs were placed and fixed in prefabricated plastic plates (dimension: 65 mm × 18 mm). Transparent adhesive tape (Scotch[TM], 3M, St. Paul, MN, USA) was used to secure the self-glazed surface from contamination. The disc was then secured with self-cure acrylic resin (DPI-RR acrylic, Mumbai, India). The surfaces were standardized using abrasive papers, progressing from a rougher to smoother grit. First, 400 grit silicon carbide paper was used to smooth the surfaces, followed by 600 grit silicon carbide papers (3M, St. Paul, MN, USA) to gradually ground the specimen surface. This was followed by a ten-second polish on a 300 rpm grinding machine (Beuhler Metaserv, Leinfelden-Echterdingen, Germany) with a water coolant. All specimens were cleansed in the running water and dried. The dried specimens were etched for 1 min with 38% phosphoric acid to clean the abrasive particles followed by ultrasonic cleaning for 1 min and then air-dried. Subsequently, the specimens were randomly divided into four groups for the different surface treatment methods. Group A was treated with 9.6% hydrofluoric acid (Pulpdent, Watertown, MA, USA) for 5 min and 30 s, rinsed with water, and then dried for 30 s with an oil-free air syringe. Group B was treated with SF 11 coarse straight fissure diamond bur (Mani, Mumbai, India) with the help of a modified dental surveyor. Group C specimens underwent laser irradiation using a $CO_2$ laser (Smart US20D, DEKA, Florence, Italy) running at a power of 13 W working at

10.6 μm in a continuous and non-contact mode. The applicator tip measured 1 mm in diameter with a length of 12 mm. The applicator tip was applied to the ceramic surfaces with a light contact for a period of 20 s in 'super mode'. Group D served as the control group and, therefore, did not receive surface treatment.

### 2.3. Observation of the Treated Surfaces of the Specimen under SEM

Specimens were sputter-coated with a gold-palladium alloy (Bal-Tec SCD 050 Sputter Coater, Bal-Tec AG, Balzers, Liechtenstein) and examined under scanning electron microscopy (JSM-5500, Jeol Ltd., Tokyo, Japan) at 15 KV in backscattered mode to determine the surface changes on the ceramic discs after treatment. The treated area of each ceramic specimen was examined and the topographical contours were viewed and photographed at original magnification ×2000.

### 2.4. Luting Ceramic Disc Specimen with Dual-Cure Resin Luting Agent and Storage

All disc surfaces were primed with a silane coupling agent (Pulpdent, Watertown, MA, USA) for 60 s and then air-dried. Afterward, two liquids of bonding agent and activator, i.e., prime & bond NT and self-cure activator, in equal quantities were combined in a mixing tray, as per the manufacturer's instruction, and applied on all of the ceramic discs. The specimens were light-cured for 10 s, according to the manufacturer's instructions. An overhead projector (OHP) sheet with a hole measuring 4 mm was placed, matching the hole, at the center of the ceramic disc. The second plate was then positioned just above the first plate, taking care that the OHP sheet should not get displaced. Machine screws were then used to tighten and stabilize both the plates together. The purpose of the 4 mm hole in the OHP sheet was to standardize the bonding area to a diameter of 4 mm. The resin cement was mixed as per the manufacturer's instructions and then applied through the 6 mm holes of the second plastic plate over the ceramic disc and light cured for 20 s from all directions. After one hour, machine screws were removed and all samples of various groups were placed in distilled water at 37 °C for 24 h in the incubator to simulate oral conditions [25,26].

### 2.5. The Shear Bond between Resin Cement and Ceramic Disc Evaluation

After 24 h in the incubator at 37 °C, all specimens were removed, rinsed, and then tested for shear bond strength (SBS) between prepared specimens of ceramic discs and resin cement on an Instron universal testing machine (Instron model 4467 tabletop load frame, 825 University Ave, Norwood, MA, USA). Tests were performed with a crosshead speed of 1 mm/min. The jaws of the Instron machine were moving in the opposite direction to test the shear bond strength between the ceramic disc specimen and resin cement. The jaws of the testing machine were continually moving in opposite directions until bond failure (debonding) took place. As soon as the bond was broken the machine automatically stopped moving and the force at bond failure in Newtons (N) was recorded for each specimen. The SBS, in megapascal (MPa), was then calculated using the formula: Force/surface area of the bracket.

### 2.6. Evaluation of Nature of Bond Failure under a Stereomicroscope

After shear bond testing of all specimens, the samples were evaluated to examine the nature of bond failure under a stereomicroscope (Wild M3B, Heerbrugg, Switzerland) at a magnification of ×12 to categorize as: (1) Cohesive, (2) Adhesive, and (3) Mixed bond failure.

*2.7. Statistical Analysis*

Data analysis was performed using SPSS v14.0 software (IBM Statistics, SPSS, Chicago, IL, USA). The basic characteristics of shear bond strength (SPS) was represented in average and standard deviation (Mean ± SD). Additionally, one-way analysis of variance (ANOVA) was performed to determine significant differences between the SBS based on the conditioning methods by implementing post hoc Tukey's test. The statistical significance was determined at $p \leq 0.05$.

### 3. Results

*3.1. Scanning Electron Microscope Examination of Surfaces*

Group A: the SEM view showed an increased irregular surface, as shown in Figure 1. The average area of irregular surface (4.83 ± 1.78 μm) acquired due to the etching effect of hydrofluoric acid on the ceramic surface was calculated using image J software (Version 1.53p 4) (Table 1).

Group B: the SEM view revealed an irregular surface is achieved by abrasion of the ceramic disc with diamond bur. The mean area (0.27 ± 0.1 μm) of irregular surface created with diamond bur was shown in Table 1. On physical observation, less irregular surface was noticed in Group B compared to Group A (Figure 2). Additionally, the irregular surface attributed by diamond treatment was significantly less when compared to hydrofluoric acid (Table 1).

Group C: the SEM view showed marginal change on the surface of the $CO_2$ laser treatment compared to Group A and B. The noticed irregularities were superficial, as shown in Figure 3. Furthermore, the mean area of irregular surface created by the $CO_2$ laser treatment was shown in Table 1.

Group D: the untreated group (control) showing few irregular surfaces may be due to phosphoric acid and porosity within the ceramic disc (Figure 4). However, the calculated mean area of irregular surface in Group D was the least compared to other analyzed groups (Table 1).

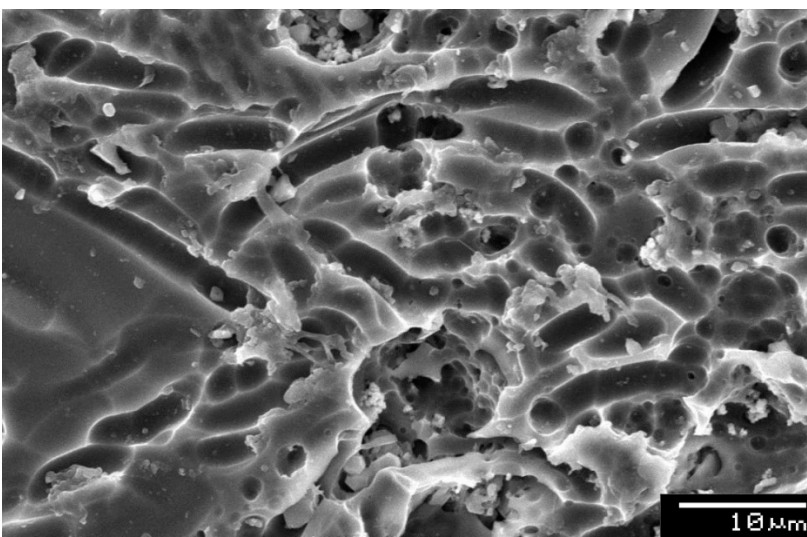

**Figure 1.** Group A: etching effect of hydrofluoric acid on the ceramic surface at ×2000 under SEM.

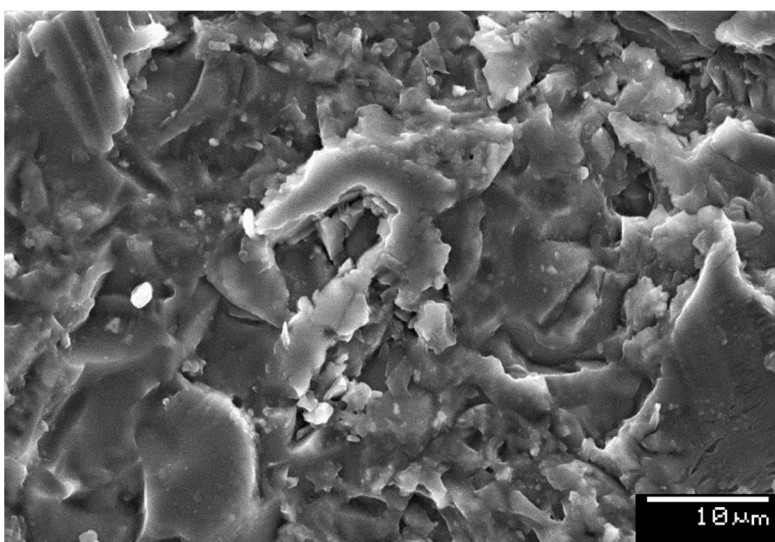

**Figure 2.** Group B: surface treatment of diamond bur on the ceramic surface at ×2000 under SEM.

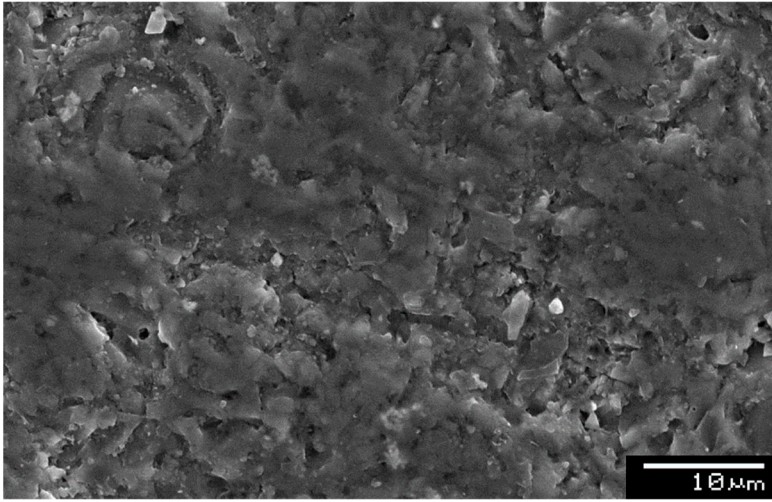

**Figure 3.** Group C: surface treatment of the $CO_2$ laser on the ceramic surface at ×2000 under SEM.

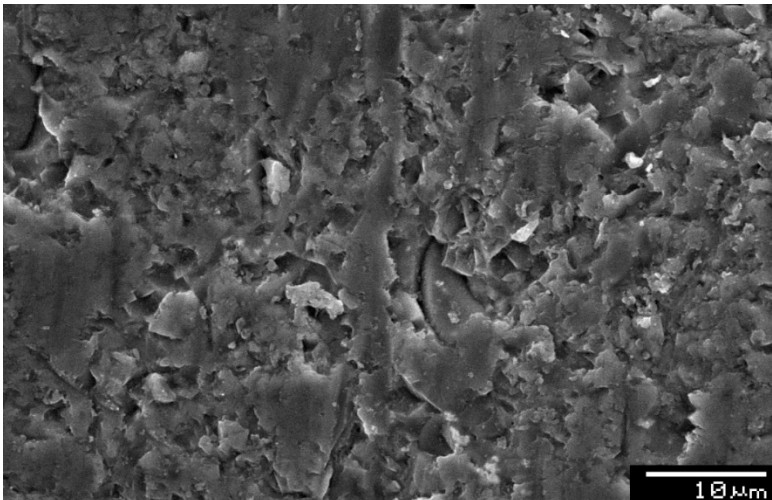

**Figure 4.** Control group (Group D): without surface treatment on the ceramic surface at ×2000 under SEM.

**Table 1.** Average irregular surface acquired after etching with hydrofluoric acid, diamond burs, and $CO_2$ laser, and control group.

| Area | Group A | Group B | Group C | Group D |
|---|---|---|---|---|
| Mean (µm) | 4.838 | 0.276 | 0.2 | 0.15 |
| SD | 1.78 | 0.1 | 0.094 | 0.045 |

*3.2. Shear Bond Strength*

The mean shear bond strength exhibited by Group D (control group) was $8.09 \pm 1.74$ MPa, which was the lowest in comparison to other specimens. The highest value of mean shear bond strength load was recorded with hydrofluoric acid (Group A—$21.40 \pm 3.57$ MPa). Diamond burs (Group B—$15.17 \pm 1.99$ MPa) showed slightly lower values of mean shear load in comparison to Group A. However, diamond bur roughening showed greater bond strength than lasers (Group C—$10.60 \pm 1.86$ MPa) (Table 2). The comparative analysis based on mean shear bond strength using ANOVA showed significant differences between groups ($p < 0.05$) (Table 3), except Group C vs. Group D.

**Table 2.** Characteristics of mean shear bond strength acquired due to hydrofluoric acid, diamond burs, and $CO_2$ laser treatment, and control.

| Group | N | Mean | Std. Dev. | 95% Confidence Interval for Mean | |
|---|---|---|---|---|---|
| | | | | Lower Bound | Upper Bound |
| Group A | 15 | 21.40 | 3.57 | 19.42 | 23.38 |
| Group B | 15 | 15.17 | 1.99 | 14.06 | 16.27 |
| Group C | 15 | 10.60 | 1.86 | 9.56 | 11.63 |
| Group D | 15 | 8.09 | 1.74 | 7.13 | 9.06 |

**Table 3.** Comparison of mean shear bond strength between the treatment and control group.

| Group | Sample Size | Statistical Significance | | | |
|---|---|---|---|---|---|
| | | Group A | Group B | Group C | Group D |
| Group A | 15 | 1 | $p \leq 0.001$ | $p \leq 0.001$ | $p \leq 0.001$ |
| Group B | 15 | $p \leq 0.001$ | 1 | $p \leq 0.001$ | $p \leq 0.001$ |
| Group C | 15 | $p \leq 0.001$ | $p \leq 0.001$ | 1 | $p = 0.52572$ |
| Group D | 15 | $p \leq 0.001$ | $p \leq 0.001$ | $p = 0.52572$ | 1 |

*p*-value $\leq 0.05$, statistically significant.

*3.3. Nature of Bond Failure*

The nature of bond failure was observed under the stereomicroscope at $40\times$ (Figures 5–7). Three types of bond failure, such as cohesive, adhesive, and mixed, were assessed for each specimen and their results were reported in Table 4. Cohesive type failure was noticed in the Group A specimen, whereas, Group B mainly showed a mixed type of bond failure. Alternatively, Group C and D showed an adhesive type failure in most of the specimens.

**Table 4.** Type of bond failure between the analyzed groups.

| Nature of Bond Failure | Group A | Group B | Group C | Group D |
|---|---|---|---|---|
| Cohesive | 15 | 03 | 00 | 00 |
| Mixed | 00 | 12 | 01 | 00 |
| Adhesive | 00 | 00 | 14 | 15 |

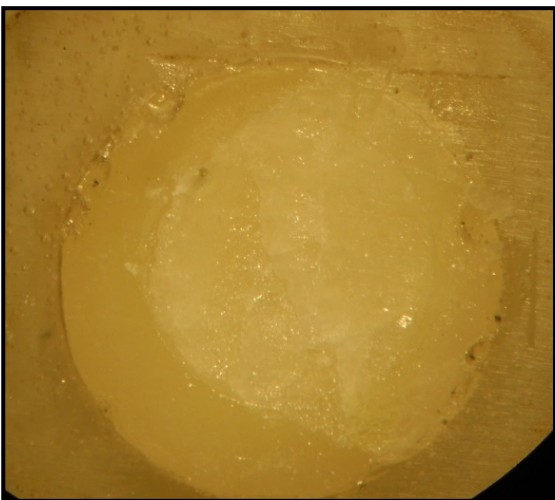

**Figure 5.** The specimen shows a cohesive type of bond failure.

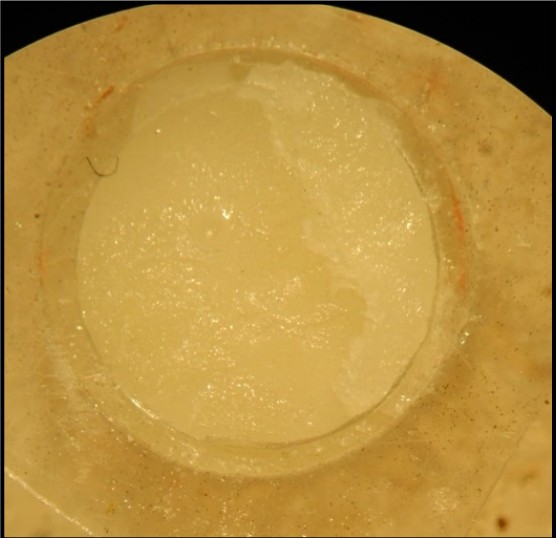

**Figure 6.** The specimen shows a mixed type of bond failure.

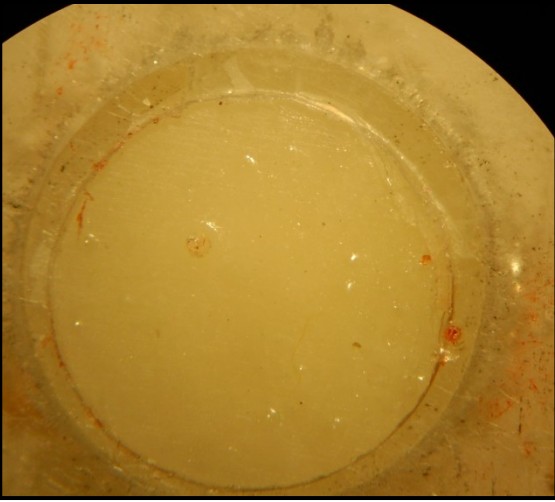

**Figure 7.** The specimen shows an adhesive type of bond failure.

## 4. Discussion

Ceramics require surface treatment for reliable bonding between the tooth and the restoration. Surface treatments are expected to modify the smooth surface by generating irregular microporosities that allow for the penetration of bonding agents for micromechanical retention. Hence, surface etching before cementation is considered as the most important step in ceramic restorations. Furthermore, the choice of surface treatment and luting agent has a potential impact on clinical success and longevity. In this study, we use three different surface treatment techniques to alter the surface characteristics of ceramics to investigate the bonding efficiency of luting agents for potential dental application. The surface characteristic after etching was assessed using SEM and the efficiency of resin bonding was determined by shear bond strength and nature bond failures. Appreciable changes in surface morphology were noted to significantly contribute to the bond strength based on the employed surface conditioning methods.

Three potential surface conditioning methods, such as (1) hydrofluoric acid, (2) diamond burs, and (3) $CO_2$ laser, were adopted to generate rough surface on the ceramic and compared with the control (untreated ceramic) [39–48]. In light of the obtained results, treatment with the hydrofluoric acid, diamond burs, and $CO_2$ laser showed significant change in the morphology of the ceramic surface compared to the control. Interestingly, the observed morphological changes with hydrofluoric acid treatment covers a wide area (4.83 ± 1.78 μm) of irregularity compared to the other two techniques, as determined from SEM image analysis using image J software. Whereas, the burs treatment showed moderate irregularity and the $CO_2$ laser showed the least irregularity compared to the control. Notably, the surface micromorphology, in terms of pores and grooves pattern, width, and depth, was considerably enormous in the hydrofluoric acid treatment, which may critically support bonding the ceramics with resin composites. Interestingly, variable sizes of grooves and pores were observed in the specimens with the hydrofluoric acid treatment. This observation supports the results of previously published studies that described the hydrofluoric acid etching appearance as tunnel-like with a honey-comb-like microretentive pattern [39–41]. Furthermore, Posritong et al. and Sato et al. showed the formation of a microstructure on glass and lithium disilicate ceramics upon hydrofluoric acid etching that produce flexural strength to the cement [40,41]. Additionally, hydrofluoric acid helps to remove the glassy matrix selectively, exposing the crystalline structures and influencing the formation of microporosities [27]. Alternatively, there are few studies that demonstrate that increased surface roughness may not always lead to an increase in bond strength [28,29]. Hydrolysis and degradation of the adhesive hybrid layer may occur in cases of inadequate moisture control that lead to failure [30,31].

Shear bond strength was assessed after cementation using the Instron universal testing machine. The hydrofluoric acid based ceramic etching showed the greatest bond strength among the analyzed groups (Table 2). The surface roughening with the diamond burs and the $CO_2$ laser group showed marginal shear bond strength compared to the untreated control group. The least bond strength was noticed in the untreated control, indicating that surface roughening on ceramics benefit efficient bonding [29–31]. This finding supports previous research in this area that shows hydrofluoric acid etching achieves greater effective bond strength compared to laser and mechanical conditioning [49–51]. Our findings are also in accord with previous studies by Shiu et al. and Akyil et al., which suggest that laser etching produces insufficient bond strength with feldspathic ceramic [7,47]. However, Hosseini et al. reported that surface treatment with lasers at a power of 1.5 to 2 W provided similar bond strengths as hydrofluoric acid etching [51]. In contrast to our findings, Barutcigil et al. examined the influence of different surface treatment methods on the shear bond strength of hybrid ceramic materials and found no difference between the techniques [52]. This disagreement might be due to the influence factors, such as acid concentration, application time, and specimen geometry.

In addition to the shear bond strength, this study evaluated the nature of bond failure acquired for the conditioning method upon cementation. Laser conditioned specimens showed an adhesive type of failure. This may be due to inadequate bonding between the resin and the ceramic surface because of a lack of roughness. It indicates that laser irradiation produces insufficient roughness for consistently reliable bond strength. The hydrofluoric acid group showed a cohesive type of failure. These results are in agreement with the findings of Moura et al. who noted hydrofluoric acid showed cohesive type failures with microshear testing [53]. This mode of failure may be related to the concentration of the acid used. Mokhtarpour et al. reported that reduced concentrations of hydrofluoric acid etchant could lead to cohesive types of failure [54]. In contrast to this, Neto et al. reported that lower concentrations of acid would result in poor bonding and lead to further adhesive failures [55]. Alternatively, increasing concentrations of hydrofluoric acid may result in loss of flexural strength. Etching time could also be an integral part of this equation with overetching ostensibly leading to non-homogeneous stress distribution during microshear bond testing.

Overall, our study revealed that the ceramic etching with hydrofluoric acid remains the most effective method, as evident from the SEM and bond strength analysis. Treating ceramic surfaces with 9.6% hydrofluoric acid produces a suitable surface for the resin bonding. However, hydrofluoric acid is caustic and hazardous to the patient and the clinician [56]. Careful and better clinical handling will be necessary to ensure that hydrofluoric acid is a gold standard that generates a rough surface on ceramic and supports micromechanical retention with resin composites. Although, this study is limited by its in vitro nature, which may not adequately simulate a true oral environment. Thus, findings of this study must be interpreted with caution when extrapolating to clinical settings with human subjects. Future studies should subject specimens to mechanical and thermal cycling to simulate the stresses and oral environment and examine the fractal geometry of various microstructures to assess the effects of various surface treatments.

## 5. Conclusions

In conclusion, our study demonstrates the effect of surface treatment with hydrofluoric acid prior to adhesive cementation that significantly contributes on shear bond strength with the resin cement compared to etching methods. Although the hydrofluoric acid was hazardous to health, careful handling may benefit the patient to achieve better ceramic restoration. Further, analyzing the effect of ceramics adhesion with different concentrations of hydrofluoric acid and relating the outcome with a benefit-risk assessment may open up new avenues in dental implantation.

**Author Contributions:** Conceptualization, S.G., M.H.D.A.W. and W.R.; methodology, B.G. and A.R.; software, B.K.M. and B.Z.; validation, S.B. (Sultan Binalrimal), V.P.V. and S.B. (Shilpa Bhandi); formal analysis, W.R. and B.K.M.; investigation, S.B. (Sultan Binalrimal) and B.G.; resources, A.T.R. and S.G.; data curation, B.Z. and S.P.; writing—original draft preparation, S.G., S.P., B.G., W.R., B.K.M. and M.H.D.A.W.; writing—review and editing, S.B. (Sultan Binalrimal), A.R., V.P.V., S.B. (Shilpa Bhandi), A.T.R. and B.Z.; visualization, A.T.R. and S.B. (Shilpa Bhandi); supervision, V.P.V. and A.R.; project administration, M.H.D.A.W. and S.P. All authors have read and agreed to the published version of the manuscript.

**Funding:** This research received no external funding.

**Institutional Review Board Statement:** Not applicable.

**Informed Consent Statement:** Not applicable.

**Data Availability Statement:** Not applicable.

**Conflicts of Interest:** The authors declare no conflict of interest.

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
