# Peer review of "The Effect of Different Surface Conditioning Techniques on the Bonding between Resin Cement & Ceramic"

_coatings, doi:10.3390/coatings12030399_

Round 1

Reviewer 1 Report

Authors evaluated the effect of three different surface conditioning techniques on the bonding between dual-cure  resin luting cemented surface treated feldspathic ceramic. The paper is worth to be published after major revision

Line

54:  Dental porcelain

86: CO2 laser

100 what kind of ceramic powder was used for disk fabricatioin

       low fusing feldspathic powder (granulation??? Chemical composition, producer )

There is no data about the Scanning electron Microsope SEM instrument, type and manufacturer Jeol,  Hitachi, Zeiss ???

There is no image in the BE mode (backscattered electrone image)

There is no EDS analysis  or EDS mapping  of the differently etched surfaces,

Why the authors did not use commercially available disks for comparison

Please add the type of Instron instrument  as well as stereomicroscope

Why you did not use the artificial saliva to demonstrate oral condition instead of distiled water??

Figure caption is always below the figure, please correct  and add more explantion of the figures.

Please add the reference

Malysa A.; et all Effect of different surface treatment methods on bond strength od dental Ceramics to dental hard tissue:a systematic revew, Molecules 202, 26,1223, doi.org/10.3390/molecules26051223

Author Response

Reviewer 1:

Authors evaluated the effect of three different surface conditioning techniques on the bonding between dual-cure  resin luting cemented surface treated feldspathic ceramic. The paper is worth to be published after major revision

Line

54:  Dental porcelain

86: CO2 laser

100 what kind of ceramic powder was used for disk fabricatioin

    Q   low fusing feldspathic powder (granulation??? Chemical composition, producer )

REPLY:

  • Dentine Porcelain Powder and liquid
  • The mixture of potassium feldspar and glass is being in the feldspathic porcelains. After incongruent melting, feldspathic porcelains contain 19 weight percentage (wt%) of leucite crystals (K2O∙Al2O3∙4SiO2)
  • CERAMCO 3 (Dentsply, Burlington, New Jersey, USA)

Q There is no data about the Scanning electron Microsope SEM instrument, type and manufacturer Jeol,  Hitachi, Zeiss ???

REPLY:

 A Scanning Electron Micro- Scope (JSM-5500, Jeol Ltd., Tokyo, Japan

Q There is no image in the BE mode (backscattered electrone image)

REPLY:

These parameters were not included in study

Q There is no EDS analysis  or EDS mapping  of the differently etched surfaces,

REPLY:

 These parameters were not included in study

Q Why the authors did not use commercially available disks for comparison

REPLY:

The present study is self-funded Study so for cost cutting it was self-prepared

 Q Please add the type of Instron instrument  as well as stereomicroscope

REPLY:

Intron model 4467 table top load frame and stereomicro- (Wild M3B, Heerbrugg, Switzerland)

Q  Why you did not use the artificial saliva to demonstrate oral condition instead of distilled water??

REPLY:

As there is no significant effect of saliva PH on bond strength so we used distilled water

 (Effect of artificial saliva and pH on shear bond strength of resin cements to zirconia-based ceramic

F Geramipanah, M Majidpour, L Sadighpour, M J Kharazi Fard) Eur J Prosthodont Restor Dent.. 2013 Mar;21(1):5-8

 Figure caption is always below the figure, please correct  and add more explantion of the figures.

 Please add the reference

Malysa A.; et all Effect of different surface treatment methods on bond strength od dental Ceramics to dental hard tissue:a systematic revew, Molecules 202, 26,1223, doi.org/10.3390/molecules26051223

REPLY:

 Reference added.

Reviewer 2 Report

  1. Dental porcelainshould be key words, and the title of the paper need be revised. Effect of different surface conditioning techniques on the bonding between resin cement & dental ceramic.  
  2. The introduaction can be improved. It would be better if some referenceswere cited. Such as, (1) Influence of ceramic surface conditioning and resin cements on microtensile bond strength to a glass ceramic,The Journal of Prosthetic Dentistry,Volume 96, Issue 6,2006,Pages 412-417,(2) Influence of different ceramic materials and surface treatments on the adhesion of Prevotella intermedia,

Journal of the Mechanical Behavior of Biomedical Materials,Volume 111,2020,104010,

  1. The supporteddata is poor for the conclusions. It would be better if the surface chemical groups were investigaed.

Author Response

Reviewer 2:

  1. Dental porcelainshould be key words, and the title of the paper need be revised. Effect of different surface conditioning techniques on the bonding between resin cement & dental ceramic.  Evaluate bond strength between resin cement and porcelain after use of different conditioning agent
  2. The introduaction can be improved. It would be better if some references were cited. Such as, (1) Influence of ceramic surface conditioning and resin cements on micro-tensile bond strength to a glass ceramic,The Journal of Prosthetic Dentistry,Volume 96, Issue 6,2006,Pages 412-417,(2) Influence of different ceramic materials and surface treatments on the adhesion of Prevotella intermedia, Journal of the Mechanical Behavior of Biomedical Materials,Volume 111,2020,104010,
  3. The supported data is poor for the conclusions. It would be better if the surface chemical groups were investigaed.

REPLY:

All the four groups were evaluated with SEM to see the effect of surface conditioning  and it was evaluated that HF was the best with feldspathic ceramic.

Introduction has been rewritten. Keywords added. References have been added. Manuscript revised.

Reviewer 3 Report

The manuscript „Effect of different surface conditioning techniques on the bonding between resin cement & ceramic“ by Gupta et al. reports on the surface morphology of ceramics discs after 3 different modification methods and their shear bond strength after luting with dual-cure resin agent. The presented work has the interesting aim of replacing hydrofluoric acid treatment with lower risk substitutions. However, the structure of the manuscript and the chosen experimental conditions make it impossible to draw useful conclusions.

My main point of criticism is that the authors state no prior hypothesis (e.g. based on literature) as to what surface structure would lead to best binding. Hence, the chosen treatment methods and specially the chosen laser paraments seem to be completely arbitrary. It is a well-known and well-studied phenomena that the laser parameters have a reproducible and strong effect on the surface morphology. Hence the authors could have gained much better insights if they would have first studied the laser parameters on the ceramic discs and chosen the best set of parameters based on the obtained morphology. In fact, the last 2 paragraphs of the discussion is the first place where the effect of surface features on the bond strength is mentioned.

I believe that the manuscript with the current findings is purely descriptive with no mechanism discussion and not suitable for publication. Also, special attention needs to be given to language, as the text is at times very hard to read.

I have a couple of specific comments that I hope will help the authors for improving the experimental work and the manuscript.

Abstract: The background does not mention the aim of this study (improvement the dental implants) so it is unclear why the work has been done.

Introduction:

-Please pay attention to support the claims with citations as the sentences are often not cited.

-Please discuss the effect of laser treatment on the surface morphology of ceramics. There are many publications that deal with this aspect.

Materials and methods:

-Fig. 1 is unnecessary as it does not show any information not stated in text.

-Was the smoothening via sanding performed with hand? How reproducible are the resulting surfaces between the 15 samples of each group?

-As stated above, the laser parameters seem to be arbitrary chosen. There is no information given about the laser spot size, the optics used, the focus determination method, the used intensity, the spot overlap, and the beam quality.

-SEM: With what device were the samples coated and what was the thickness used?

What device was the SEM and what detector has been used?

-Shear bond measurements: the authors state that the machine “give a recording automatically in the form of a graph and figure” what is this graph and what information can be obtained from it? With what force were the samples pulled?

- Evaluation of nature of bond failure under a stereomicroscope: Please provide images and show how the nature of the bond failure can be evaluated using a microscope. No details about the experimental procedure for achieving the conclusions is given here.

Results:

The structure of the manuscript in the results section does not fit the materials and methods section. Were the SEM images obtained after the breakage? If not please move them to the beginning and discuss them.

Table1: As stated above, please provide image proof as how the optical micrographs allow drawing conclusions about the nature of failure.

Table 2: it could be advantages to show this table as a graph and show the statistics on the graph. It is very unclear what groups showed statistically significant differences.

The SEM images: The features on the samples are very different. Some show cracks while others show rod like shapes. Which ones would strengthen the bonding? Please remove the information bar from the SEM images.

Discussion: Again, as the hypothesis is not supported by the investigation of the effect of morphology on the bonding, the discussion also gives very little information as to why hydrofluoric acid treatment is the best. There is no mechanism discussed and the discussion is rather a review of the literature.

Conclusion:

Since the manuscript results are descriptive without elucidating any mechanisms, the conclusion is rather a summary.

Author Response

Reviewer 3:

The manuscript „Effect of different surface conditioning techniques on the bonding between resin cement & ceramic“ by Gupta et al. reports on the surface morphology of ceramics discs after 3 different modification methods and their shear bond strength after luting with dual-cure resin agent. The presented work has the interesting aim of replacing hydrofluoric acid treatment with lower risk substitutions. However, the structure of the manuscript and the chosen experimental conditions make it impossible to draw useful conclusions.

My main point of criticism is that the authors state no prior hypothesis (e.g. based on literature) as to what surface structure would lead to best binding. Hence, the chosen treatment methods and specially the chosen laser paraments seem to be completely arbitrary. It is a well-known and well-studied phenomena that the laser parameters have a reproducible and strong effect on the surface morphology. Hence the authors could have gained much better insights if they would have first studied the laser parameters on the ceramic discs and chosen the best set of parameters based on the obtained morphology. In fact, the last 2 paragraphs of the discussion is the first place where the effect of surface features on the bond strength is mentioned.

REPLY:

Co2 details added

The specimens were treated by using CO2 laser (Smart US-20D, DEKA, Firenze, Italy) working at 10.6μm. 13W in continuous and non-contact mode. The application tip’s diameter was 1mm and its length was 12mm. Moving up and down, ceramic  surfaces were processed with the application tip in slight contact.super mode for 20 seconds

The carbon dioxide laser (CO2) is commonly used intraorally  in soft tissue and hard tissue applications [1].  Dental porcelain approximately  absorbs CO2 laser wavelength, CO2 laser is very suitable for the surface treatment of ceramic materials [2]. CO2 laser etching may represent an effective method for conditioning zirconia surfaces, enhancing micromechanical retention and improving the bond strength [3]. There is Conchoidal tears result from surface warming, because of heat initiation of surfaces of ceramic by focusing CO2 laser. Which may leads mechanical success between resin composite and ceramics retention [4-6].

  1. Pick RM, Colvard MD. Current status of lasers in soft tissue dental surgery. J Periodontol. 1993;64:589–602.
  2. Ural Ç, Külünk T, Külünk Åž, Kurt M. The effect of laser treatment on bonding between zirconia ceramic surface and resin cement. Acta Odontol Scand. 2010;68:354–9.
  3. Akyil MS, Uzun IH, Bayindir F. Bond strength of resin cement to yttrium-stabilized tetragonal zirconia ceramic treated with air abrasion, silica coating, and laser irradiation. Photomed Laser Surg. 2010;28:801–8.
  4. Ersu B, Yuzugullu B, Ruya Yazici A, Canay S. Surface roughness and bond strengths of glass-infiltrated alumina ceramics prepared using various surface treatments. J Dent. 2009;37:848–56.
  5. Chen JR, Oka K, Kawano T, Goto T, Ichikawa T. Carbon dioxide laser application enhances the effect of silane primer on the shear bond strength between porcelain and composite resin. Dent Mater J. 2010;29:731–7.
  6. Akova T, Yoldes O, Toroglu MS, Uysal H. Porcelain surface treatment by laser for bracket-porcelain bonding. Am J Orthod Dentofac Orthop. 2005;128:630–7.

I believe that the manuscript with the current findings is purely descriptive with no mechanism discussion and not suitable for publication. Also, special attention needs to be given to language, as the text is at times very hard to read.

I have a couple of specific comments that I hope will help the authors for improving the experimental work and the manuscript.

Abstract: The background does not mention the aim of this study (improvement the dental implants) so it is unclear why the work has been done.

Introduction:

-Please pay attention to support the claims with citations as the sentences are often not cited.

-Please discuss the effect of laser treatment on the surface morphology of ceramics. There are many publications that deal with this aspect.

Materials and methods:

-Fig. 1 is unnecessary as it does not show any information not stated in text.

-Was the smoothening via sanding performed with hand? How reproducible are the resulting surfaces between the 15 samples of each group?

REPLY:

 In order to standardize surfaces, 400 - grit silicon carbide paper followed by 600- grid silicon carbide papers (English abrasives, English abrasives Ltd. England) were gradually grounded using water coolant on a 300rpm grinding machine for 10s (Beuhler  Metaserv, Germany), all specimens were cleansed in the running water. All dried specimens were etched for 1 min with 38% phosphoric acid to clean the abrasive particles followed by ultrasonically cleaned for 1min and then air-dried. Subsequently, specimens were randomly divided into four groups for the following different surface treatment methods.

-SEM: With what device were the samples coated and what was the thickness used?

REPLY:

Each group were Gold-Palladium Alloy sputter-coated (Bal-Tec SCD 050 Sputter Coater, Bal-Tec AG, Liechtenstein)

What device was the SEM and what detector has been used?

REPLY:

 Observed With A Scanning Electron Micro- Scope (JSM-5500, Jeol Ltd., Tokyo, Japan

-Shear bond measurements: the authors state that the machine “give a recording automatically in the form of a graph and figure” what is this graph and what information can be obtained from it? With what force were the samples pulled?

REPLY:

Tests were performed with a crosshead speed of 1mm/min. Jaws of the Instron machine were moving in the opposite direction to test the shear bond strength between ceramic disc specimen and resin cement. Jaws of testing machine continually moving up in opposite direction till the bond failure took place. As soon as the bond was broken machine automatically stopped moving and give a recording automatically in the form of a readings in megapascals (MPa).

- Evaluation of nature of bond failure under a stereomicroscope: Please provide images and show how the nature of the bond failure can be evaluated using a microscope. No details about the experimental procedure for achieving the conclusions is given here.

The Specimen Shows Cohesive Type Of Bond Failure       

The Specimen Shows Mixed Type Of Bond Failure

The Specimen Shows Adhesive Type Of Bond Failure

Results:

The structure of the manuscript in the results section does not fit the materials and methods section. Were the SEM images obtained after the breakage? If not please move them to the beginning and discuss them.

REPLY:

Images of SEM were not after breakage. They are after particular conditioning treatment

Table1: As stated above, please provide image proof as how the optical micrographs allow drawing conclusions about the nature of failure.

REPLY:

As mention above

Table 2: it could be advantages to show this table as a graph and show the statistics on the graph. It is very unclear what groups showed statistically significant differences.

 The SEM images: The features on the samples are very different. Some show cracks while others show rod like shapes. Which ones would strengthen the bonding? Please remove the information bar from the SEM images.

Discussion: Again, as the hypothesis is not supported by the investigation of the effect of morphology on the bonding, the discussion also gives very little information as to why hydrofluoric acid treatment is the best. There is no mechanism discussed and the discussion is rather a review of the literature.

Conclusion:

Since the manuscript results are descriptive without elucidating any mechanisms, the conclusion is rather a summary.

REPLY:

We thank the reviewer for taking time in scientifically guiding us and indicating the shortcomings of our manuscript. We, the authors, have reworked and revised the entire manuscript based on the reviewer’s points and hope it is satisfactory. Hypothesis has been clearly mentioned. Relevant citations added. Introduction, discussion and conclusion have been rewritten.

Round 2

Reviewer 1 Report

The Authors revised their manuscript of  evaluation the effect of three different surface conditioning techniques on the bonding between dual-cure  resin luting cemented surface treated feldspathic ceramic. The paper is now worth to be published.

Author Response

Reviewer 1:

Comment: The Authors revised their manuscript of   evaluation the effect of three different surface conditioning techniques on the bonding between dual-cur resin luting cemented surface treated feldspathic ceramic. The paper is now worth to be published.

Response: Thanks for considering the manuscript for publication

Reviewer 3 Report

I thank the authors for answering some of the comments raised before. However, as most of my comments are not answered and the rest are answered only partially, I have no choice but to again ask the authors to carefully revise the manuscript. I believe the results would be interesting for the readers if the authors take the time and thoroughly work on the comments. Below please find the comments that were not answered and the new questions based on the give answers.

REPLY:

Co2 details added

The specimens were treated by using CO2 laser (Smart US-20D, DEKA, Firenze, Italy) working at 10.6μm. 13W in continuous and non-contact mode. The application tip’s diameter was 1mm and its length was 12mm. Moving up and down, ceramic  surfaces were processed with the application tip in slight contact.super mode for 20 seconds

The carbon dioxide laser (CO2) is commonly used intraorally  in soft tissue and hard tissue applications [1].  Dental porcelain approximately  absorbs CO2 laser wavelength, CO2 laser is very suitable for the surface treatment of ceramic materials [2]. CO2 laser etching may represent an effective method for conditioning zirconia surfaces, enhancing micromechanical retention and improving the bond strength [3]. There is Conchoidal tears result from surface warming, because of heat initiation of surfaces of ceramic by focusing CO2 laser. Which may leads mechanical success between resin composite and ceramics retention [4-6].

  1. Pick RM, Colvard MD. Current status of lasers in soft tissue dental surgery. J Periodontol. 1993;64:589–602.
  2. Ural Ç, Külünk T, Külünk Åž, Kurt M. The effect of laser treatment on bonding between zirconia ceramic surface and resin cement. Acta Odontol Scand. 2010;68:354–9.
  3. Akyil MS, Uzun IH, Bayindir F. Bond strength of resin cement to yttrium-stabilized tetragonal zirconia ceramic treated with air abrasion, silica coating, and laser irradiation. Photomed Laser Surg. 2010;28:801–8.
  4. Ersu B, Yuzugullu B, Ruya Yazici A, Canay S. Surface roughness and bond strengths of glass-infiltrated alumina ceramics prepared using various surface treatments. J Dent. 2009;37:848–56.
  5. Chen JR, Oka K, Kawano T, Goto T, Ichikawa T. Carbon dioxide laser application enhances the effect of silane primer on the shear bond strength between porcelain and composite resin. Dent Mater J. 2010;29:731–7.
  6. Akova T, Yoldes O, Toroglu MS, Uysal H. Porcelain surface treatment by laser for bracket-porcelain bonding. Am J Orthod Dentofac Orthop. 2005;128:630–7.

Question: I thank the author for adding the details of the laser parameters. However, these discussions and citations are not added to the manuscript. Your future readers may also have the same questions as I do and adding the citations and discussion in the manuscript is necessary. What type of roughness (cracks, pores, rods,…) and with what quantification did the authors want to achieve?

Abstract: The background does not mention the aim of this study (improvement the dental implants) so it is unclear why the work has been done.

Question: improvement the dental implants is still not added to the abstract

Materials and methods:

-Fig. 1 is unnecessary as it does not show any information not stated in text.

Question: What is the reason for keeping Fig 1. It does not add any information.

-Was the smoothening via sanding performed with hand? How reproducible are the resulting surfaces between the 15 samples of each group?

REPLY:

 In order to standardize surfaces, 400 - grit silicon carbide paper followed by 600- grid silicon carbide papers (English abrasives, English abrasives Ltd. England) were gradually grounded using water coolant on a 300rpm grinding machine for 10s (Beuhler  Metaserv, Germany), all specimens were cleansed in the running water. All dried specimens were etched for 1 min with 38% phosphoric acid to clean the abrasive particles followed by ultrasonically cleaned for 1min and then air-dried. Subsequently, specimens were randomly divided into four groups for the following different surface treatment methods.

Question: How reproducible are the resulting surfaces between the 15 samples of each group? Are the surfaces quantified?

-SEM: With what device were the samples coated and what was the thickness used?

REPLY:

Each group were Gold-Palladium Alloy sputter-coated (Bal-Tec SCD 050 Sputter Coater, Bal-Tec AG, Liechtenstein)

Question: what was the thickness of the coating used?

What device was the SEM and what detector has been used?

REPLY:

 Observed With A Scanning Electron Micro- Scope (JSM-5500, Jeol Ltd., Tokyo, Japan

Question: what detector has been used? Secondary? Backscattered?

-Shear bond measurements: the authors state that the machine “give a recording automatically in the form of a graph and figure” what is this graph and what information can be obtained from it? With what force were the samples pulled?

REPLY:

Tests were performed with a crosshead speed of 1mm/min. Jaws of the Instron machine were moving in the opposite direction to test the shear bond strength between ceramic disc specimen and resin cement. Jaws of testing machine continually moving up in opposite direction till the bond failure took place. As soon as the bond was broken machine automatically stopped moving and give a recording automatically in the form of a readings in megapascals (MPa).

Question: Is this recording in MPa the breaking stress? Please provide more information

- Evaluation of nature of bond failure under a stereomicroscope: Please provide images and show how the nature of the bond failure can be evaluated using a microscope. No details about the experimental procedure for achieving the conclusions is given here.

The Specimen Shows Cohesive Type Of Bond Failure

The Specimen Shows Adhesive Type Of Bond Failure

Question: From the quality of the images, I still cannot see how the nature of the bond failure can be evaluated. Can the author please explain? Please remove the line numbers from the images.

Results:

The structure of the manuscript in the results section does not fit the materials and methods section. Were the SEM images obtained after the breakage? If not please move them to the beginning and discuss them.

REPLY:

Images of SEM were not after breakage. They are after particular conditioning treatment

Question: Putting the SEM images after the shear bond strength section makes it look like they were made after the breakage, as mentioned in the first round, please change the structure and move them to the section before the bond breakage so it is clear that the SEM images are from pristine samples.

Table1: As stated above, please provide image proof as how the optical micrographs allow drawing conclusions about the nature of failure.

REPLY:

As mention above

Question: From the quality of the images I still cannot see how the nature of the bond failure can be evaluated. Can the author please explain?

Table 2: it could be advantages to show this table as a graph and show the statistics on the graph. It is very unclear what groups showed statistically significant differences.

Question: This comment is still open

 The SEM images: The features on the samples are very different. Some show cracks while others show rod like shapes. Which ones would strengthen the bonding?

Question: This comment is still open

Please remove the information bar from the SEM images.

Question: Thank you for removing the information bar, however you also removed the scale bars! Please provide scale bars.

Discussion: Again, as the hypothesis is not supported by the investigation of the effect of morphology on the bonding, the discussion also gives very little information as to why hydrofluoric acid treatment is the best. There is no mechanism discussed and the discussion is rather a review of the literature.

Question: This comment is still open

Conclusion:

Since the manuscript results are descriptive without elucidating any mechanisms, the conclusion is rather a summary.

Question: This comment is still open

Author Response

Reviewer 3:

I thank the authors for answering some of the comments raised before. However, as most of my comments are not answered and the rest are answered only partially, I have no choice but to again ask the authors to carefully revise the manuscript. I believe the results would be interesting for the readers if the authors take the time and thoroughly work on the comments. Below please find the comments that were not answered and the new questions based on the give answers.

Resopnse:

Co2 details added

The specimens were treated by using CO2 laser (Smart US-20D, DEKA, Firenze, Italy) working at 10.6μm. 13W in continuous and non-contact mode. The application tip’s diameter was 1mm and its length was 12mm. Moving up and down, ceramic  surfaces were processed with the application tip in slight contact.super mode for 20 seconds

The carbon dioxide laser (CO2) is commonly used intraorally in soft tissue and hard tissue applications [1].  Dental porcelain approximately  absorbs CO2 laser wavelength, CO2 laser is very suitable for the surface treatment of ceramic materials [2]. CO2 laser etching may represent an effective method for conditioning zirconia surfaces, enhancing micromechanical retention and improving the bond strength [3]. There is Conchoidal tears result from surface warming, because of heat initiation of surfaces of ceramic by focusing CO2 laser. Which may leads mechanical success between resin composite and ceramics retention [4-6].

  1. Pick RM, Colvard MD. Current status of lasers in soft tissue dental surgery. J Periodontol. 1993;64:589–602.
  2. Ural Ç, Külünk T, Külünk Åž, Kurt M. The effect of laser treatment on bonding between zirconia ceramic surface and resin cement. Acta Odontol Scand. 2010;68:354–9.
  3. Akyil MS, Uzun IH, Bayindir F. Bond strength of resin cement to yttrium-stabilized tetragonal zirconia ceramic treated with air abrasion, silica coating, and laser irradiation. Photomed Laser Surg. 2010;28:801–8.
  4. Ersu B, Yuzugullu B, Ruya Yazici A, Canay S. Surface roughness and bond strengths of glass-infiltrated alumina ceramics prepared using various surface treatments. J Dent. 2009;37:848–56.
  5. Chen JR, Oka K, Kawano T, Goto T, Ichikawa T. Carbon dioxide laser application enhances the effect of silane primer on the shear bond strength between porcelain and composite resin. Dent Mater J. 2010;29:731–7.
  6. Akova T, Yoldes O, Toroglu MS, Uysal H. Porcelain surface treatment by laser for bracket-porcelain bonding. Am J Orthod Dentofac Orthop. 2005;128:630–7.

Question1: I thank the author for adding the details of the laser parameters. However, these discussions and citations are not added to the manuscript. Your future readers may also have the same questions as I do and adding the citations and discussion in the manuscript is necessary. What type of roughness (cracks, pores, rods,…) and with what quantification did the authors want to achieve?

Response: Our sincere apology for not adding the information on “CO2 laser”. In the current version of the manuscript, we have added the necessary details (along with the references).  Also, regarding type of roughness, we noticed different features on each group (as you mentioned in question 15). To be noted the Diamond bur and HFA treatment showed several morphological feature (not identical). Whereas, Co2 laser and control looks similar. More roughness was achieved by HFA was clearly established from the SEM analysis (Table.1 in the manuscript), which contribute efficiently for bonding (see Table2A in manuscript).  Simultaneously, the shear bond testconfirm the HFA treated ceramic with more roughness contribute efficient high affinity with the cement (Table.2A).

===========================================================================

Question2: improvement the dental implants is still not added to the abstract

Abstract: The background does not mention the aim of this study (improvement the dental implants) so it is unclear why the work has been done.

Response: The aim of this study now included in the abstract section of the revised manuscript.

=========================================================================

 Materials and methods:

-Fig. 1 is unnecessary as it does not show any information not stated in text.

Question3: What is the reason for keeping Fig 1. It does not add any information.

Response: We have removed the Figure.1 from the revised manuscript.

===========================================================================

Question 4: -Was the smoothening via sanding performed with hand? How reproducible are the resulting surfaces between the 15 samples of each group?

Response: Detailed information on the smoothing methodology is now added to the revised manuscript (Refer: Page# 3, last paragraph). All specimens underwent similar procedure for smoothing. Hence, the surface smoothing will be highly reproducible. Additionally, the specimens were randomly divided into four groups and subjected to different surface conditioning techniques.

 ============================================================================

Question5: SEM: With what device were the samples coated and what was the thickness used?

Response: Each sample was Gold-Palladium Alloy sputter-coated (Bal-Tec SCD 050 Sputter Coater, Bal-Tec AG, Liechtenstein) for conductivity at nanometer thickness.   

============================================================================

Question 6: What device was the SEM and what detector has been used?  Secondary? Backscattered?

Response: We used Scanning Electron Micro- Scope (JSM-5500, Jeol Ltd., Tokyo, Japan ) in Backscattered mode (Refer: Page 4, 2nd paragraph).

============================================================================

Question 7: -Shear bond measurements: the authors state that the machine “give a recording automatically in the form of a graph and figure” what is this graph and what information can be obtained from it? With what force were the samples pulled?

Response: The crosshead speed of 1mm/min. The shear bond strength (SBS) was assessed using the Instron machine on the treated ceramic from the cement. The Bond strength was calculated using the formula given in the manuscript at is displayed in MPa (mean ± SD) (Refer: page 4, last paragraph).

=============================================================================

Question 8: Is this recording in MPa the breaking stress? Please provide more information

- Evaluation of nature of bond failure under a stereomicroscope: Please provide images and show how the nature of the bond failure can be evaluated using a microscope. No details about the experimental procedure for achieving the conclusions is given here.

The Specimen Shows Cohesive Type Of Bond Failure

The Specimen Shows Adhesive Type Of Bond Failure

Response: The bond strength for each group and presented in MPa (mean ± SD). The image recoding for the nature of bond failure under a stereomicroscope added to the manuscript. In the current manuscript, we have added the about the experimental procedure for arriving the conclusions on bond failure. ==================================================================

Question 9: From the quality of the images, I still cannot see how the nature of the bond failure can be evaluated. Can the author please explain? Please remove the line numbers from the images.

Response:  SEM image provides roughness on the ceramic. However, it does not demonstrate the nature of bond failure.  In our study, the nature bond failure was assessed using the stereomicroscope and the outcome was shown in the Figure5-7 and Table.3.

 As suggested the numbers present in the SEM images were removed and numerical scale was maintained.

============================================================================

Question10 : The structure of the manuscript in the results section does not fit the materials and methods section. Were the SEM images obtained after the breakage? If not please move them to the beginning and discuss them.

Response:  SEM images were taken before breakage (Figure 1-4).  Considering the reviewer suggestion, we alter the manuscript structure for better understanding.

 ===================================================

Question 11: Putting the SEM images after the shear bond strength section makes it look like they were made after the breakage, as mentioned in the first round, please change the structure and move them to the section before the bond breakage so it is clear that the SEM images are from pristine samples.

Response: Thanks for the reviewer suggestion. We have made the changes in the revised manuscript.

==================================================================

Question 12: Table1: As stated above, please provide image proof as how the optical micrographs allow drawing conclusions about the nature of failure.

Response: We have added the images relating bond failure in the manuscript (Figure5-7).

=============================================================================

Question 13: From the quality of the images I still cannot see how the nature of the bond failure can be evaluated. Can the author please explain?

Response:  We use SEM and its images provide roughness on the ceramic. However, it does not demonstrate the nature of bond failure.  In our study, the nature bond failure was assessed using the stereomicroscope and the outcome was shown in the Figure 5-7 & Table.3.

=========================================================================

Question 14: Table 2: it could be advantages to show this table as a graph and show the statistics on the graph. It is very unclear what groups showed statistically significant differences.

Question: This comment is still open

 Response: Statistical significance is now clearly defined in revised manuscript (Table.)  

=========================================================================

Question 15: The SEM images: The features on the samples are very different. Some show cracks while others show rod like shapes. Which ones would strengthen the bonding?

Question: This comment is still open

 Response: The SEM images we noticed different type of roughness features, in each group. To be noted the BUR and HFA showed more irregular feature (not identical). Whereas,Co2 laser and control looks similar. Much roughness achieved by HFA which established from the SEM image using imageJ software (Table.1), which contributes efficient bonding.  The bonding strength was assessed using Instron universal testing machine which  confirms the bonding efficiency (Table.2A).

=========================================================================

Please remove the information bar from the SEM images.

Question 16: Thank you for removing the information bar, however you also removed the scale bars! Please provide scale bars.

Response: We have now added scale bar to the SEM images

 ========================================================================

Discussion: Again, as the hypothesis is not supported by the investigation of the effect of morphology on the bonding, the discussion also gives very little information as to why hydrofluoric acid treatment is the best. There is no mechanism discussed and the discussion is rather a review of the literature.

Question 17: This comment is still open

Response: The discussion part was rewritten with the evidence from our results along with the supporting evidence from previous literature.

=========================================================================

Conclusion:

Since the manuscript results are descriptive without elucidating any mechanisms, the conclusion is rather a summary.

Question 18: This comment is still open

Response: The conclusion now modified based on our investigation and result outcome.

========================================================================

Round 3

Reviewer 3 Report

I thank the authors for answering all the comments in the second round and suggest publishing after very minor language editing.